# Simulation and Machine Learning Assessment of P-Glycoprotein Pharmacology in the Blood–Brain Barrier: Inhibition and Substrate Transport

**DOI:** 10.3390/ijms26189050

**Published:** 2025-09-17

**Authors:** Christian Jorgensen, Elizabeth Oliphant, Milly Barker, Eduardo López Martínez, Saaihasamreen Thulasi, Holly Prior, Ben William Franey, Charley Gregory, Jerry Oluwasegun, Anjalee Ajay, Roger R. Draheim

**Affiliations:** 1School of Medicine, Pharmacy and Biomedical Sciences, Faculty of Science & Health, University of Portsmouth, Portsmouth PO1 2DT, UK; up2115139@myport.ac.uk (E.O.); milly.barker@myport.ac.uk (M.B.); saaihasamreen.thulasi@myport.ac.uk (S.T.); up2102739@myport.ac.uk (H.P.); up2096876@myport.ac.uk (B.W.F.); up2047688@myport.ac.uk (C.G.); jerry.oluwasegun@myport.ac.uk (J.O.); anjalee.ajay@myport.ac.uk (A.A.); roger.draheim@port.ac.uk (R.R.D.); 2Department of Chemistry, Aarhus University, Langelandsgade 140, 8000 Aarhus, Denmark; 3Laboratory of Computational Biophysics of Macromolecules, School of Chemical Sciences, Meritorious Autonomous University of Puebla (BUAP), Puebla 72570, Mexico

**Keywords:** blood–brain barrier, efflux pumps, P-glycoprotein, molecular dynamics, cryo-EM, substrates, inhibitors, PDB database mining, coarse-grained modeling, Boltz-2

## Abstract

We explored the pharmacology of the P-glycoprotein (P-gp) efflux pump and its role in multidrug resistance. We used Protein Data Bank (PDB) database mining and the artificial intelligence (AI) model Boltz-2.1.1, developed for simultaneous structure and affinity prediction, to explore the multimeric nature of recent P-gp inhibitors. We construct a MARTINI coarse-grained (CG) force field description of P-gp embedded in a model of the endothelial blood–brain barrier. We found that recent P-gp inhibitors have been captured in either monomeric, dimeric, or trimeric states. Our CG model demonstrates the ability of P-gp substrates to permeate and transition across the BBB bilayer. We report a multimodal binding model of P-gp inhibition in which later generations of inhibitors are found in dimeric and trimeric states. We report analyses of P-gp substrates that point to an extended binding surface that explains how P-gp can bind over 300 substrates non-selectively. Our coarse-grained model of substrate permeation into membranes expressing P-gp shows benchmarking similarities to prior atomistic models and provide new insights on far longer timescales.

## 1. Introduction

The blood–brain barrier (BBB), formed primarily by brain microvascular endothelial cells (BMECs) and surrounded by mural cells and astrocytic end feet, regulates the transport of nutrients, molecules, and cells into and out of the brain. BMECs form tight junctions to restrict paracellular transport, while various transporters and efflux pumps facilitate and regulate transcellular transport [1]. Despite its protective function, the BBB poses a significant obstacle to the treatment of many central nervous system (CNS) disorders by limiting drug delivery to the brain.

A major contributor to this limited permeability is P-glycoprotein (P-gp; Figure 1a), an ATP-binding cassette (ABC) transporter associated with multidrug resistance (MDR) [2]. Polarized to the luminal surface of BMECs, P-gp functions as an ATP-dependent efflux pump that transports substrates back into circulation that have passively diffused into the cells [3]. Efflux pumps like P-gp act as a secondary line of protection for the brain and have numerous molecular substrates, contributing to multidrug resistance (MDR). Enhancing CNS drug delivery thus requires strategies that either inhibit P-gp-mediated efflux or MDR. Although over 300 compounds have been identified as potential P-gp substrates [4], no strategy has yet proven successful in clinically inhibiting P-gp [5], underscoring the need for deeper mechanistic understanding.

P-gp is a ~170 kDa protein composed of two pseudo-symmetric halves that span two transmembrane domains (TMDs) and two nucleotide-binding domains (NBDs) (Figure 1a). Each TMD is made up of six transmembrane helices. During the efflux cycle, P-gp adopts two distinct conformations: an inward-facing state, which opens intracellularly, and an outward-facing state, which opens extracellularly. In the inward-facing state, the protein forms an inverted V shape, where the two structural halves delineate an inner cavity through which substrates gain access to the transporter core. For efflux back into circulation (Figure 1b), substrates must access this inner volume [6]. The aperture angle of the protein (θ) is defined by the spacing (d) between the NBDs at the ends of each arm, and both the aperture angle and NBD spacing change during efflux as conformational rearrangements occur. Structural studies have reported that the NBD spacing in the inward-facing state ranges from 40 to 60 Å [7,8,9,10], though a recent study of human P-gp reported a narrower spacing of approximately 30 Å [10]. In reviews of P-gp structures, significant structural heterogeneity has been observed, including discrepancies between aperture angles observed in cryo-electron microscopy (cryo-EM) structures and those derived from X-ray crystallography [6]. For these reasons, it is crucial to mention that a large number of P-gp cryo-EM structures have been released [1,2]. While a lot of work has gone into understanding the conformational transitions of P-gp [11,12,13,14,15,16,17,18], we will not cover in this work.

In vitro and in vivo studies of P-gp function offer an avenue to understanding P-gp efflux kinetics and the influence of P-gp on Caco-2 cell permeability [1,19,20,21,22]. However, these methods are often limited in their ability to offer mechanistic insight into the steps associated with efflux. To address this limitation, molecular dynamic (MD) simulations are increasingly employed to probe conformational transitions and substrate interactions beyond the reach of current experimental techniques. Coarse-grained (CG) simulations have become an attractive option that bypasses some of the sampling limitations to classical MD [23,24], with recent MARTINI 3 developments and successes suggesting that the field of CG simulations is reaching a mature stage [23,24,25,26].

In addition to atomistic simulations, the in silico repertoire includes a rising number of AI and machine learning (ML) tools applied to predict novel P-gp substrates and inhibitors. Such studies have used a broad range of techniques, and we highlight just a few. Supervised ML models such as support vector machines (SVMs) [27,28,29] have been used since the early 2000s to predict novel P-gp substrates, and with great success. As a matter of fact, the often-quoted number of 300 P-gp substrates originates from a paper that utilized SVMs [29]. Other studies have used convolutional neural networks [30], such as the novel ligand-based convolutional neural network (NLCNN) model that classifies P-gp substrates to a prediction accuracy of 80% based on a curated dataset of 197 P-gp substrates, by integrating molecular docking and ligand-based deep learning methods for further predictive improvement [30]. Similarly, graph neural networks (GNNs), such as the AttentiveFP model [31] that is trained on a dataset of 1995 drug molecules (1202 substrates, 793 nonsubstrates) and has a receiver operating characteristic area under the curve (ROC-AUC) of 0.848 and an accuracy of 0.815, have been used.

In this study, we used PDB database mining, AI tools, and a coarse-grained model to glean novel insights into the binding of both inhibitors and substrates of P-gp. We propose a multimodal binding model for P-gp inhibition that holds the promise of better understanding the complex problem of P-gp inhibition for preventing multidrug resistance.

## 2. Results

### 2.1. P-gp Inhibitor Pharmacology: Multimodality of Inhibition

By PDB database mining, we present evidence of P-gp modulation that includes examples with monomeric (*n* = 1), dimeric (*n* = 2), and trimeric (*n* = 3) inhibitor bound. Firstly, P-gp was captured with a monomer of QZ-Leu (PDB id 4Q9K; X-ray, 3.80 Å; Figure 2a). The binding pose shown in Figure 2a employs a van der Waals representation of the inhibitor and also depicts the chemical structure of QZ-Leu. Secondly, the inhibition of P-gp by a dimer of tariquidar was identified (PDB id 7A6E; cryo-EM, 3.60 Å; Figure 2b). Thirdly, the inhibition of P-gp by a trimeric bundle of elacridar (PDB id 8Y6I; cryo-EM, 2.54 Å; Figure 2c) is depicted in Figure 2c. Figure 2d illustrates the calculated center-of-mass (COM) distance to the apical top, defined in Figure 1 as F335 and F336, which increased as the inhibitor modality changed from monomer to trimer. This is due to the COM shifting downwards towards the NBD lobes as the trimeric assembly occupies more space in the central cavity (Figure 2). In Figure 2e–g, we depict the molecular interactions between P-gp and each inhibitor molecule. Figure 2e depicts the inhibition of P-gp by a QZ-Leu monomer (PDB id 4Q9K; X-ray, 3.80 Å), while Figure 2f illustrates the P-gp binding of a tariquidar dimer (PDB id 7A6E; cryo-EM, 3.60 Å). Finally, Figure 2g shows P-gp inhibition by a trimer of elacridar (PDB id 8Y6I; cryo-EM, 2.54 Å).

Further analysis of P-gp inhibitor geometry is given in the Appendix A. Appendix A depicts the position of the inhibitors within the central binding cavity ABC. In Appendix A we examine the hinge angle geometry of structures with 1st generation inhibitors QZ-X (X = Val, Leu) compared to later generation inhibitors tariquidar and elacridar. In Appendix A, we examine the internal positioning of dimeric and trimeric inhibitors, which shows the multimeric inhibitors extend laterally outwards as opposed to vertically upward or downward from the binding site of a monomeric inhibitor.

### 2.2. P-Glycoprotein Substrate Pharmacology: The Handling of Chemotherapeutics and Environmental Pollutants from PDB Database Analysis

Using PDB database mining, we sought to characterize the broad range of novel P-gp substrate transport structures. The structures we considered are shown in Figure 3. Firstly, we considered P-gp bound with the substrate and chemotherapeutic Taxol (PDB id 6QEX; cryo-EM, 3.60 Å; Figure 3a) [10], secondly, P-gp bound with the substrate and environmental polluter bromophenyl BDE100 (PDB id 6UJN; X-ray, 3.98 Å; Figure 3b) [35], and thirdly, P-gp bound with the substrate and the cystic fibrosis transmembrane conductance regulator (CFTR) potentiator ivacaftor (PDB id 7OTG; cryo-EM, 5.40 Å; Figure 3c) [36]. The average distance of each substrate to the apex (F335, F336) is shown in Figure 3d. In Figure 3e,f, we depict the binding site of Taxol, BDE100, and ivacaftor in P-gp. We found that the Taxol and BDE100 were positioned at the same distance to the apical point defined by F335 and F336. On the other hand, ivacaftor was located further down the inner binding site. For purposes of the distance comparison, we built a homology model based on the 6UJN template of murine P-gp with BDE100 (see Section 4).

### 2.3. Assessing IC_50_ of P-gp Inhibitors and Substrates with AI

Using the AI-based tool Boltz-2 [37] we calculated the affinity probability and the predicted IC_50_ for each of the three inhibitors (QZ-Leu, tariquidar, elacridar) and the three substrates (Taxol, BDE100, ivacaftor). The results are presented in Table 1. The predicted template modeling (pTM) score considers the overall fold accuracy of a single protein chain [37]. The predicted inter-chain TM (ipTM) score assesses the inter-chain interactions in multimeric models [37]. The confidence score showcases the overall model quality. The average predicted local distance difference test (pLDDT) is the confidence metric per residue [37] and reflect the local structural reliability. The affinity probability estimates the likelihood that a predicted protein–ligand complex corresponds to a true, energetically favorable binding interaction [37]. The IC_50_ denotes the half-maximal inhibitory concentration, and is defined as the modulator concentration required to inhibit 50% of target activity, with a lower value denoting a higher modulator potency. The predicted negative log_10_ of IC_50_ (pIC_50_) correlates with the potency of inhibition, with a higher value predicting a greater inhibitory effect.

Figure 4 demonstrates the output from the Boltz-2 prediction for P-gp substrate and inhibitor binding sites, specifically for inhibitors QZ-Leu (Figure 4a), tariquidar (Figure 4b), and elacridar (Figure 4c), and for substrates Taxol (Figure 4d), BDE100 (Figure 4e), and ivacaftor (Figure 4f). It is apparent that the performance of Boltz-2 is best for monomeric binding, while Boltz-2 is currently unable to predict dimeric and trimeric poses, which are being predicted to occur from cryo-EM, because the algorithm tends to approximate these molecules as monomeric entities, hence failing to capture their full structural complexity.

Figure 5 shows a comparison of the binding poses predicted by Boltz-2 (orange) to those from X-ray crystallography or cryo-EM (red) for the P-gp inhibitor QZ-Leu, the P-gp substrate BDE100, and the P-gp substrate ivacaftor. This comparison suggests that Boltz-2 is a valid tool for predicting P-gp substrate and inhibitor binding locations. The results demonstrate that Boltz-2 effectively identifies binding sites for both substrates and inhibitors with high accuracy, as the predicted poses were consistent with those obtained from cryo-EM or X-ray, either as native or as homology models, thereby supporting the reliability of this new engine.

### 2.4. Developing a P-Glycoprotein Coarse-Grained Model to Explore Substrate Interactions

Here, we report a coarse-grained (CG) model of P-gp embedded in the BMEC membrane. We compared the model of partitioning of the P-gp substrate rhodamine 123 to previously published atomistic models of the BBB [38,39] and P-gp embedded in a model of the endothelial BBB [6]. The composition used for the endothelial BBB membrane has been reported elsewhere [38], and comprises ~30% cholesterol, ~19% sphingomyelin, and the remainder (~51%) of phospholipids. The protein system was prepared by the Martinize tool of MARTINI [40] using an atomistic equilibrated coordinate of human inward-facing (IF) P-glycoprotein (PDB ID 6QEX [10]). A protein–bilayer system was prepared with the INSANE application of MARTINI [40] using a coarse-grained (CG) topology description under the MARTINI 3.0.b.3.2 coarse-grained force field [41], specifically with a four-to-one mapping that typically represent four atomistic particles as one bead. An elastic network model was added to the protein description [42] to preserve the secondary structure. A protein–bilayer system was prepared using a 322-lipid bilayer distributed as in Table 1, comprising 30% cholesterol, 6% POPE, 19% POSM, 8% SLPC, 8% PAPS, 15% PAPE, 4% POPC, 8% PAPC, and 2% SAPI. The simulation box was solvated with MARTINI water beads. Na^+^ and Cl^−^ ions were added to the system to reach the physiological conditions of 150 mM NaCl. We chose a typical substrate, rhodamine 123, which we have studied previously [6], as a representative example of a general P-gp substrate. The results are depicted in Figure 6. In Figure 6a–c, we depict our analysis of the coarse-grained model of the apo P-gp trajectory (50 μs) using MARTINI with an elastic network model [42]. The root-mean-square deviation (RMSD ± σ) of P-gp relative to the initial conformation yielded an average value of 6.1 ± 0.4 Å for replica 1 and 6.1 ± 0.3 Å for replica 2, similar to that calculated for ubiquitin using MARTINI and MARTINI with an elastic network [43]. In Figure 6b–d, we depict the partitioning of 20 CG rhodamine 123 substrate using a MARTINI representation (red), showing that the partition of substrate into the BMEC bilayer reaches over 90% within the first microsecond. In Figure 6e, we depict the percentage of partitioning of CG rhodamine 123 to atomistic (AA) rhodamine 123 (blue) from our previous work [6]. This percentage represents the number of substrate molecules that have partitioned into the membrane, and shows that the models agree quantitatively on the percentage of substrate partitioning, which provides model validity.

## 3. Discussion

### 3.1. P-gp Inhibition Exhibits Multimodal Character

The inhibitors analyzed in this study support a multimodal binding model. Based on PDB data mining (Figure 2a–c), the inhibitors were observed to interact with multiple pockets and occupy increasingly larger volumes within the central binding cavity (CBC) of P-gp as the binding mode changed from monomeric to multimeric. This spatial expansion allows inhibitors to interact with a broader array of residues across the transmembrane domains, resulting in a wider contact surface compared to a monomeric mode of inhibition. This could contribute to an increased inhibitory effect by physically obstructing substrate access and preventing conformational transitions. However, this also correlates with a reduced specificity for the canonical substrate-binding pocket, particularly near the apex of the central binding cavity, also supported by increasing distances from F335 and F336 residues in Figure 2d. We found the distance of the inhibitor from the P-gp apex increased with progression from monomeric forms (QZ-Leu) to multimeric forms (tariquidar and elacridar) occurs, and this change was accompanied by a larger standard deviation (σ) in ligand positioning across models. This suggestion that the apical peak (W335/W336) plays a role in the anchoring the extracellular end of the TM helix, as has been reported in other membrane-spanning proteins [45,46]. In summary, monomeric inhibitors such as QZ-Leu are located more distally in the inner binding cavity.

We wish to credit the work Nosol et al. for some of the original cryo-EM models of dimeric inhibitor poses [32]. They originally proposed the idea that P-gp preferentially binds globular compounds, such as the aromatic ring systems contained by tariquidar, elacridar, or zosuquidar. They proposed two distinct mechanisms of P-gp inhibition: (1) inhibition by a single molecule X in the CBC, when the molecule X is present in high concentration, and (2) inhibition by a pair (dimer) of molecules, in which the first molecule X sits as in (1), while a second dimer molecule X’ extends outwards transversally. They furthermore suggested a concentration gradient between such globular molecules acting as substrates at low concentrations to inhibiting P-gp at higher concentrations. This is not new, as there has long been debate in the field regarding the status of tariquidar as either a substrate or an inhibitor [47]. The work of Weidner et al. [47] addressed this debate, positing tariquidar as a classical inhibitor. The work of Nosol [32] allows for a dual substrate-inhibitory character of such globular compounds, thus appeasing both sides of the debate.

Furthermore, our data reveal that the inhibitors occupy different binding sites and interact with distinct sets of amino acid residues, as shown in Figure 2e–g. QZ-Leu binds near the canonical substrate-binding pocket, making direct contact with M986, a residue also observed contacting substrate Taxol in Figure 3e. This overlap supports a competitive and targeted binding pattern. In contrast, the dimeric tariquidar and trimeric elacridar complexes span multiple sites within the central cavity, extending away from the apex and interacting with more peripheral residues, including Y130. This broader engagement may reduce their ability to tightly occupy or compete for the precise substrate-binding site while still achieving inhibition by blocking substrate access and conformational stabilization of the inward-facing state.

### 3.2. P-gp Substrate Pharmacology: The Handling of Chemotherapeutics and Environmental Pollutants from PDB Database Analysis Shows Diversity in Character

Over 300 P-gp substrates have been identified [4], many of which are chemotherapeutics and other essential precision medicine tools, and this has motivated an intense search for a P-gp inhibitor, which to this day has not resulted in late-stage clinical trial success [6]. One way to improve this search is to better understand the nuances among substrates, in particular the interactions of P-gp across diverse substrate classes such as environmental pollutants (e.g., BDE100) and chemotherapeutic agents (e.g., Taxol).

In Figure 3a–d we depict the substantial variability in substrate orientation and proximity to the apical residues F335 and F336 (apex definition; Figure 1a). Compared to the inhibitors shown in Figure 2, substrates generally maintain a greater average distance from these residues (Figure 3d. One could infer that this points to differences in the mechanism substrate between transport and inhibition, but such discussions have been held elsewhere vigorously [48]. We note that Taxol (PDB ID: 6QEX) and BDE100 (PDB ID: 6UJN) show similar spatial distances from the apex (F335–F336), implying overlapping binding positions despite their differing chemical classes. In contrast, the CFTR potentiator ivacaftor (PDB ID: 7OTG) binds at a lower point in the inner cavity, displaying a larger center-of-mass distance from these residues. This deviation could point to a subtle differentiated mode of interaction, which has been explored elsewhere [49]. It is our general opinion, however, that the P-gp inner binding region should be considered a single, extended binding surface, as we have argued elsewhere [6]. This argument succinctly explains how P-gp can bind over 300 different substrates non-selectively. Figure 3 also reveals the presence of conserved interaction patterns between substrates and inhibitors within the P-gp-binding pocket, in particular shared molecular contacts of residue E875 with both the inhibitor elacridar (Figure 2c) and the substrate ivacaftor (Figure 3c). These shared contacts point to common recognition motifs potentially crucial to P-gp function. Residue Q347 emerged as another key binding site, interacting with the substrate ivacaftor (Figure 3f), inhibitor tariquidar (Figure 2f), and substrate Taxol (Figure 3e). This convergence suggests Q347 may play a versatile role in the promiscuous binding behavior characteristic of P-gp. Similarly, residue M986 interacts with both Taxol (Figure 3e) and QZ-Leu (Figure 2e), further supporting the concept of shared binding determinants between substrates and inhibitors.

These conserved contact points represent promising targets for rational drug design, offering new strategies to modulate P-gp activity by exploiting common recognition motifs shared by structurally diverse compounds.

### 3.3. Boltz-2 AI Suggests That Later Generation Inhibitors Bind More Efficiently

To rigorously assess the reliability of structural and binding predictions, Boltz-2 employs a suite of confidence metrics tailored to both single-chain and multi-chain protein models. The predicted TM score (pTM) quantifies confidence in single-chain structural prediction, representing the algorithm’s estimate of the traditional TM score that would result from superimposing the predicted structure onto the unknown native conformation. For multi-chain assemblies, the interface predicted TM score (ipTM) serves as a complementary metric, evaluating the accuracy of inter-chain packing and interactions by considering only inter-chain residue pairs.

Binding affinity is assessed through two complementary metrics. The binary binding affinity score estimates the probability of ligand–target interaction, while the predicted pIC_50_—expressed as −log_10_(IC_50_ in molar units)—provides a quantitative measure of binding potential, with higher values indicating stronger predicted affinity. Additionally, the predicted IC_50_ value reflects the half-maximal inhibitory concentration, where lower values denote greater binding strength.

As shown in Table 1, all the predicted inhibitor poses had high confidence scores. Elacridar stands out with the highest pTM score (0.781), indicating the greatest confidence in single-chain structural prediction, and the highest ipTM score (0.933), reflecting superior inter-chain packing accuracy. In contrast, tariquidar demonstrates the highest predicted pIC_50_ value (7.243) and the lowest predicted IC_50_ value (57.1 nM), suggesting both the strongest binding potential and greatest inhibitory strength. Based on the combined metrics, tariquidar emerges as the most confident and potent inhibitor according to Boltz-2 predictions.

Similarly, all substrates listed in Table 1 show robust confidence scores. BDE100 exhibits the highest pTM score (0.792), while ivacaftor achieves the highest ipTM score (0.915) and binding affinity probability (0.663). Taxol displays the highest predicted pIC_50_ (6.928) and IC_50_ (118.0 nM) values among substrates, indicating strong binding potential and inhibitory capacity. Based on the combined metrics, Taxol emerges as the most confident and potent substrate according to Boltz-2 predictions.

A trend emerges whereby the earlier first-generation inhibitors such as QZ-Leu [7] are associated with less potent P-gp inhibition and larger IC_50_ values. For QZ-Leu and the associated QZ family of inhibitors, Szewczyk et al. estimated IC_50_ values in vitro to increase with the size of the side chain, namely QZ-Val (IC_50_ = 1.7 µM), QZ-Leu (IC_50_ = 5.4 µM) and QZ-Phe (IC_50_ = 24 µM) [7]. We proceeded to compare these experimental values to those estimated by Boltz-2, namely QZ-Val (IC_50_ = 1.4 µM), QZ-Leu (IC_50_ = 0.27 µM) and QZ-Phe (IC_50_ = 0.26 µM) (Appendix A). The ratio of Boltz-2 IC_50_ values to the experimental IC_50_ (IC_50,Boltz_/IC_50,invitro_) were as follows: QZ-Val: ~1 QZ-Leu ~20 QZ-Phe ~92.

For the tariquidar dimer (Figure 2b), Boltz-2 predicts the IC_50_ of tariquidar in the nanomolar regime (IC_50_ = 57.1 nM). This is in agreement with the experimental characterization of tariquidar as a potent third-generation P-gp inhibitor (IC_50_ = 43 nM [50]) that binds strongly to P-gp (*K*_d_
*=* 5.1 nM) [47].

### 3.4. P-gp Coarse-Grained Model

Our coarse-grained model of substrate permeation into P-gp shows benchmarking similarities to prior atomistic models [6], but this model provides new insights on far longer timescales. Our CG model is able to observe the diffusion of P-gp in the membrane over a 50 μs timescale. The protein RMSD (μ ± σ) over 50 μs for replica 1 was found to be 6.1 ± 0.4 Å, and for replica 2 it was 6.1 ± 0.3 Å, which was similar to that calculated for ubiquitin using MARTINI and MARTINI with an elastic network [43]. In Figure 6b–d, we depict the partitioning of 20 CG rhodamine 123 substrate using a MARTINI representation (red), showing that partition of substrate into the BMEC bilayer reaches over 90% within the first microsecond. In Figure 6e, we depict the percentage partitioning of CG rhodamine 123 to atomistic (AA) rhodamine 123 (blue) from our previous work [6], showing that the models agree quantitatively on the percentage of substrate partitioning, which yields model validity.

### 3.5. Conclusions

In conclusion, in this work we report a multimodal binding model of P-gp inhibition in which P-gp can be inhibited by compounds bound as monomer, dimers, or trimers. This is a trend that has emerged in the Protein Data Bank in recent years, and both raises many questions and bears important consequences for the next generation of P-gp inhibitors to combat multidrug resistance to chemotherapeutics. From machine learning calculations with Boltz-2, we report analysis of P-gp inhibitors and substrates and show that machine learning can predict the binding pose to a high degree of correspondence with high-resolution X-ray or cryo-EM models. We report analysis of P-gp substrates using machine learning together with our newly developed P-gp-BBB coarse-grained model, which point to an extended binding region that explains how P-gp can bind over 300 substrates non-selectively.

The P-gp multimodal model holds promise to help advance our clinical knowledge of P-gp inhibition. When considering that all prior clinical trials on P-gp inhibition have been classed as unsuccessful at the late stage [51], this challenge thus becomes more urgent. Our P-gp coarse-grained model could be used for future studies to better understand how P-gp substrates interact with the protein at long timescales, although the main bottleneck is the non-trivial nature of generating small-molecule simulation parameters.

## 4. Materials and Methods

### 4.1. P-gp Human Homology Models

P-gp homology models have been described elsewhere in detail [52]. The following human P-gp homology models were constructed in this work.

(1) A human homology model from the murine inward-facing P-gp template (ABCB1; PDB ID: 4Q9K; resolution: 3.8 Å; sequence identity: 87%; UniProt: P08183) from the Uniprot server (European Bioinformatics Institute EMBL-EBI & SIB Swiss Institute of Bioinformatics, https://www.uniprot.org/, accessed on 1 January 2025). Target–template alignment was performed using ProMod version 3 via the SWISS-MODEL webserver (Computational Structural Biology Group at the SIB Swiss Institute of Bioinformatics at the Biozentrum, University of Basel, https://swissmodel.expasy.org, accessed on 1 January 2025) [53]. The resulting alignment and verification data are shown in Appendix A, respectively. Model quality was assessed using the global and per-residue QMEAN scoring function [54].

(2) To compare substrate-bound P-gp structures based on a human template, BDE100 (ABCB1; murine P-gp; PDB ID: 6UJN; resolution: 3.98 Å) and ivacaftor (ABCB1; murine P-gp; PDB ID: 7OTG; resolution: 5.40 Å) were remodeled using SWISS-MODEL. This approach ensured consistency with the human Taxol-bound structure (PDB ID: 6QEX). The modeling process involved four key steps: (i) template identification: UniProt sequence P08183 (human P-gp) was selected. (ii) Target–template alignment: alignment was conducted between UniProt sequence P08183 (human P-gp) and P21447 (mouse P-gp). (iii) Model building: homology models of the ligand-bound structures were generated. (iv) Model quality evaluation: see Appendix A for relevant assessment metrics, and Appendix A for the alignment assessment. The SWISS-MODEL workflow did not retain the original ligand-binding sites in the remodeled structures. Therefore, the ligands were manually reinserted into the homology model post-construction to preserve their docking orientation. STAMP (structural alignment of multiple proteins) structural alignment of template and the resulting homology model was achieved with MultiSeq [55] in VMD 1.9.1 (Beckman Institute for Advanced Science and Technology & National Institutes of Health & National Science Foundation; Physics, Computer Science, and Biophysics at University of Illinois at Urbana-Champaign, Urbana-Champaign, IL) [34] prior to reinsertion.

### 4.2. Protein–Ligand Co-Folding Using Boltz-2

Structural predictions of protein–ligand complexes, as well as estimated binding affinity and IC_50_ values, were performed using Boltz-2.1.1, a machine learning-based model developed for simultaneous structure and affinity prediction [37] on the Rowan Scientific web server (Rowan Scientific Corporation, Boston MA, https://rowansci.com/, accessed on 30 May 2025) using the protein–ligand co-folding tool, which utilizes the Boltz-2 model. Protein sequences were retrieved from UniProt (ID: P21447) [56], and ligand structures were provided in SMILES format, which were sourced from PubChem [57] with CIDs 441276, 154083, and 16220172, respectively. A default multiple sequence alignment (MSA) server was used for generating evolutionary profiles automatically. This step was handled internally by the backend of Boltz-2.

### 4.3. Coarse-Grained Protein–Membrane Model Setup

The protein system was prepared by the Martinize tool of MARTINI (Martini Force Field Initiative, Groningen, NL) [40] using an atomistic equilibrated coordinate of human inward-face (IF) P-glycoprotein (PDB ID 6QEX [10]). A protein–bilayer system was prepared with the INSANE application of MARTINI [40] using a coarse-grained (CG) topology description under the MARTINI 3.0.b.3.2 coarse-grained force field [41], specifically with a four-to-one mapping that typically represents 4 atomistic particles as one bead. An elastic network model was added to the protein description [42] to preserve the secondary structure. The final bilayer size was 322 lipids (Table 2). The box was solvated with MARTINI water beads (each bead represents 4 water molecules). Na^+^ and Cl^−^ ions were added to the system to reach the physiological conditions of 150 mM NaCl.

### 4.4. Coarse-Grained Molecular Dynamics

All simulations (including all-atom) were run on the GROMACS 2021 (Royal Institute of Technology and Uppsala University, Sweden) simulation package [58]. Following system setup, an equilibration procedure consisting of 50 ns using the Berendsen barostat [59] and v-rescale thermostat [60] for pressure and temperature coupling, respectively, was conducted. Simulations were run at 300 K and 1 bar pressure. An atmospheric pressure of 1 bar was maintained using Berendsen pressure coupling with compressibility κ = 4.5 · 10^−5^ bar^−1^ and a time constant τ_P_ = 4 ps. The LINCS algorithm [61] was used to constrain bond lengths with a timestep of 20 fs. The long-range electrostatic interactions were calculated with the particle mesh Ewald (PME) method. During the equilibration process, the protein backbone beads were constrained. The 50 ns of equilibration was continued with a 50 μs production run for apo P-gp. A second simulation with 20 CG rhodamine 123 substrate molecules was added to the equilibrated box and run for 12 μs.

### 4.5. Software

All chemical renderings were prepared with VMD 1.9.1 [34]. Simulations were run on the GROMACS 2021 simulation package [58]. Final figures were prepared with Inkscape 1.3.2.

## Figures and Tables

**Figure 1 ijms-26-09050-f001:**
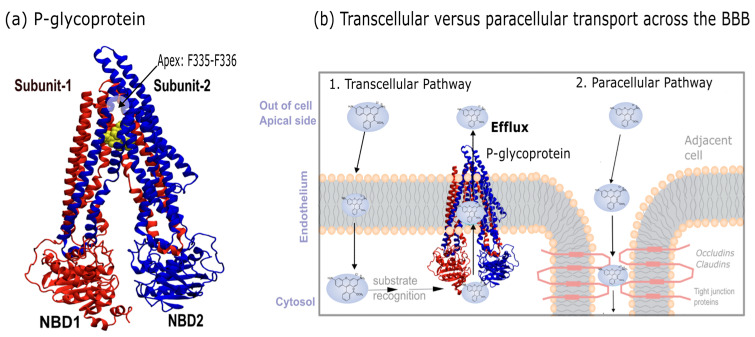
P-gp and its role in efflux of chemotherapeutics and other important therapeutics across the cell membrane. (**a**) P-glycoprotein consists of two pseudo-symmetric halves encoded into a single polypeptide. Each half consists of six transmembrane helices and one cytosolic nucleotide-binding domain (NBD), along with interconnecting loops and short helices. The two structural halves enclose a central cavity. An invariant apex point is made up of residues F335 and F336. (**b**) Schematic illustration showing transport processes across the endothelial BBB. The first pathway is transcellular diffusion, in which molecules passively diffuse across the membrane, but can be transported back into circulation by efflux pumps if they are a substrate of such pumps. The second pathway is the paracellular pathway, in which molecules exploit the cell–cell space for transport. Due to the presence of tight junction proteins such as occludins and claudins, this pathway is mostly inaccessible. As such, for most small lipophilic molecules, passive diffusion is the only pathway into the brain. Color code: NBD1 is depicted in red, and NBD2 is depicted in blue.

**Figure 2 ijms-26-09050-f002:**
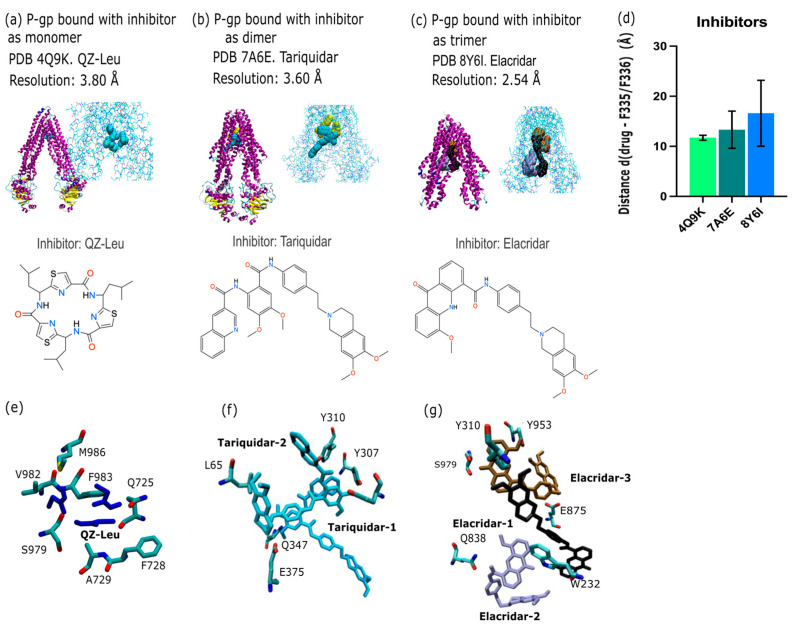
Multimodality of P-gp inhibition arranged by state of oligomerization. (**a**) Inhibition of P-gp by a monomer of QZ-Leu (PDB id 4Q9K; X-ray, 3.80 Å) [7], with the binding pose shown in van der Waals representation and associated chemical structure depicted. (**b**) Inhibition of P-gp by a dimer of tariquidar (PDB id 7A6E; cryo-EM, 3.60 Å) [32]. (**c**) Inhibition of P-gp by a trimer of elacridar (PDB id 8Y6I; cryo-EM, 2.54 Å) [33]. This structure possesses an unresolved NBD–NBD lobe region. (**d**) Average distance between the center of mass (COM) of each inhibitor moiety and the COM residue F335 and residue F336. Due to the presence of an uneven number of inhibitors between (**a**–**c**), the distance of each inhibitor is calculated to F335 and the procedure is repeated for F336. The error bar is the standard deviation (σ) of each individual distance obtained per structure. (**e**,**f**) Binding site analysis of recent P-gp inhibitors in the PDB. (**e**) Inhibition of P-gp by a monomer of QZ-Leu (PDB id 4Q9K; X-ray, 3.80 Å) [7], with the binding pose shown in licorice representation. (**f**) Inhibition of P-gp by a dimer of tariquidar (PDB id 7A6E; cryo-EM, 3.60 Å) [32]. (**g**) Inhibition of P-gp by a trimer of elacridar (PDB id 8Y6I; cryo-EM, 2.54 Å) [33]. Figure panels were rendered with VMD 1.9.1 [34]. Color coding: P-gp depicted with secondary structure coloring scheme from VMD, inhibitors shown in cyan (monomer), yellow (dimer) and as trimer a coloring scheme consisting of black, purple and brown was utilized.

**Figure 3 ijms-26-09050-f003:**
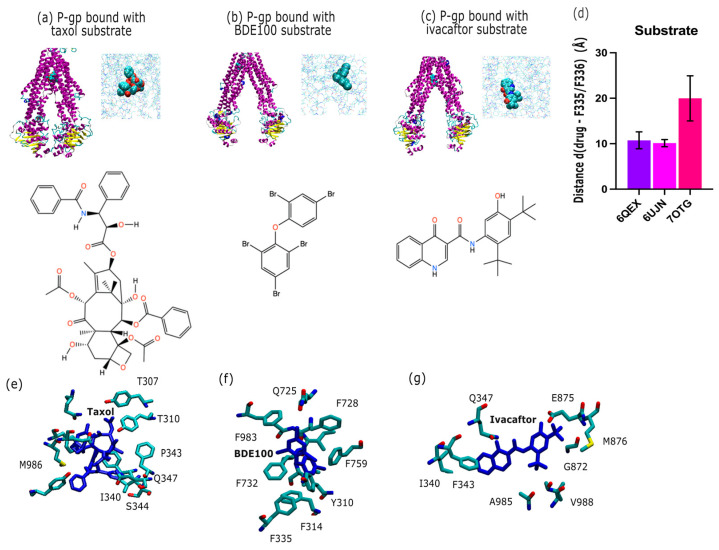
Overview of the binding site of novel P-gp substrate conformations available in the PDB. (**a**) P-gp bound with the substrate and chemotherapeutic Taxol (PDB id 6QEX; cryo-EM, 3.60 Å) [10]. (**b**) P-gp bound with the substrate and environmental polluter bromophenyl BDE100 (PDB id 6UJN; X-ray, 3.98 Å) [35]. (**c**) P-gp bound with the substrate and CFTR potentiator ivacaftor (PDB id 7OTG; cryo-EM, 5.40 Å) [36]. (**d**) Average distance between the center of mass (COM) of each substrate moiety and the COM residue F335 and residue F336. To follow the same standard as for the inhibitors in Figure 2, the distance of each substrate was calculated to F335 and the procedure repeated for F336. The error bar is the standard deviation (σ) of each individual distance obtained per structure. (**e**–**g**) Overview of the binding site of recent P-gp substrates of chemotherapeutic or environmental concern available in the PDB. (**e**) P-gp bound with the substrate and chemotherapeutic Taxol (PDB id 6QEX; cryo-EM, 3.60 Å) [10], with the binding pose shown in licorice. (**f**) P-gp bound with the substrate and environmental polluter bromophenyl BDE100 (PDB id 6UJN; X-ray, 3.98 Å) [35]. (**g**) P-gp bound with the substrate and CFTR potentiator ivacaftor (PDB id 7OTG; cryo-EM, 5.40 Å) [36]. Figure panels were rendered with VMD 1.9.1 [34]. Color coding: P-gp depicted with secondary structure coloring scheme from VMD, substrates shown in cyan. Substrates are shown in dark blue in (**e**,**f**).

**Figure 4 ijms-26-09050-f004:**
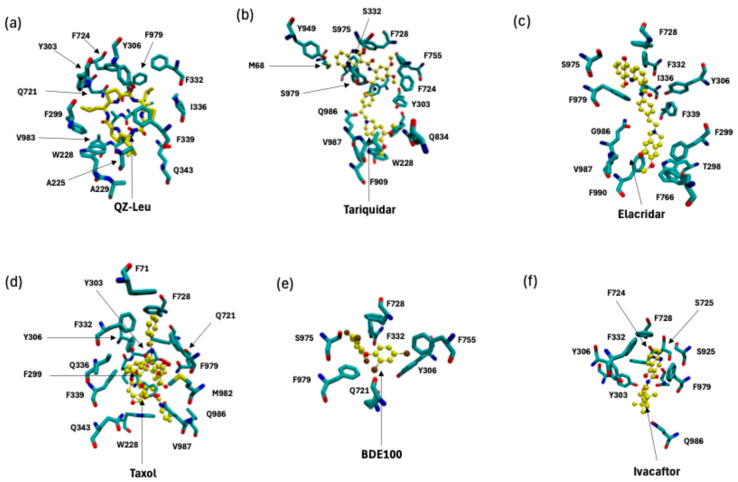
Binding poses predicted by the AI tool Boltz-2 for (**a**–**c**) P-gp inhibitors and (**d**–**f**) P-gp substrates. (**a**) P-gp inhibition by QZ-Leu, (**b**) P-gp inhibition by tariquidar, (**c**) P-gp inhibition by elacridar. (**d**–**f**) Overview of the binding site of recent P-gp substrates of chemotherapeutic or environmental concern available in the PDB. (**d**) P-gp bound with the substrate and chemotherapeutic Taxol (PDB id 6QEX; cryo-EM, 3.60 Å) [10], with the binding pose shown in CPK rendering style, with the environment shown in licorice style. (**e**) P-gp bound with the substrate and environmental polluter bromophenyl BDE100 (PDB id 6UJN; X-ray, 3.98 Å) [35]. (**f**) P-gp bound with the substrate and CFTR potentiator ivacaftor (PDB id 7OTG; cryo-EM, 5.40 Å) [36]. Figure panels were rendered with VMD 1.9.1 [34]. Color coding: inhibitor or substrate depicted in yellow.

**Figure 5 ijms-26-09050-f005:**
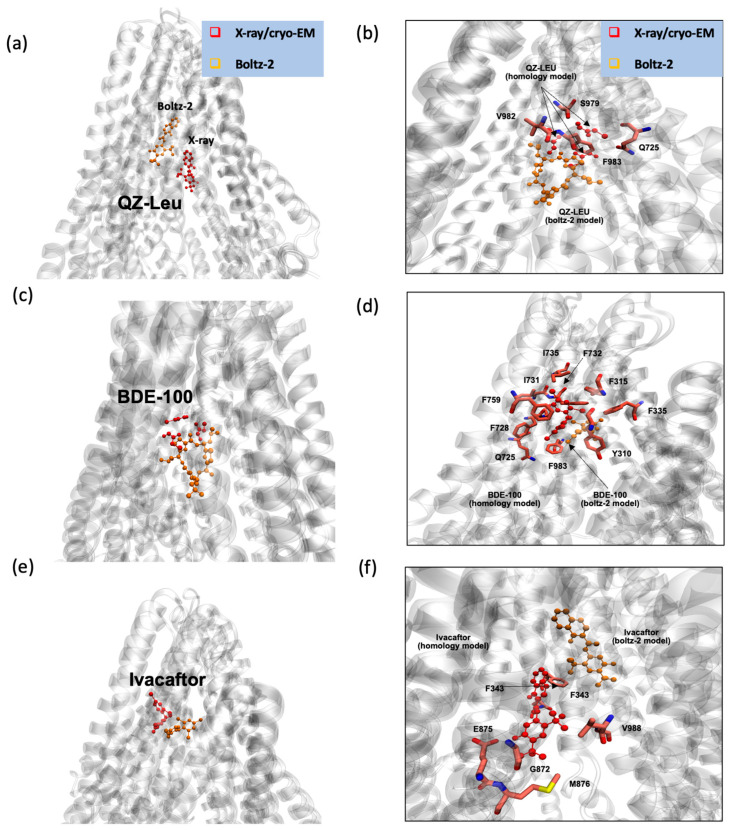
Comparing the binding poses predicted by the AI tool Boltz-2 (orange) to those from X-ray crystallography or cryo-EM (red) for the P-gp inhibitor QZ-Leu, the P-gp substrate BDE100, and the P-gp substrate ivacaftor. (**a**,**b**) P-gp inhibition by QZ-Leu (PDB id 4Q9K; X-ray, 3.80 Å) [7]. (**c**,**d**) P-gp bound with the substrate and environmental polluter bromophenyl BDE100 (PDB id 6UJN; X-ray, 3.98 Å) [35]. (**e**,**f**) P-gp bound with the substrate and CFTR potentiator ivacaftor (PDB id 7OTG; cryo-EM, 5.40 Å) [36]. This comparison suggests that Boltz-2 is a valid tool for predicting P-gp substrate and inhibitor binding locations. Figure panels were rendered with VMD 1.9.1 [34]. Color code: pose from Boltz-2 in orange, pose from X-ray/cryo-EM in red.

**Figure 6 ijms-26-09050-f006:**
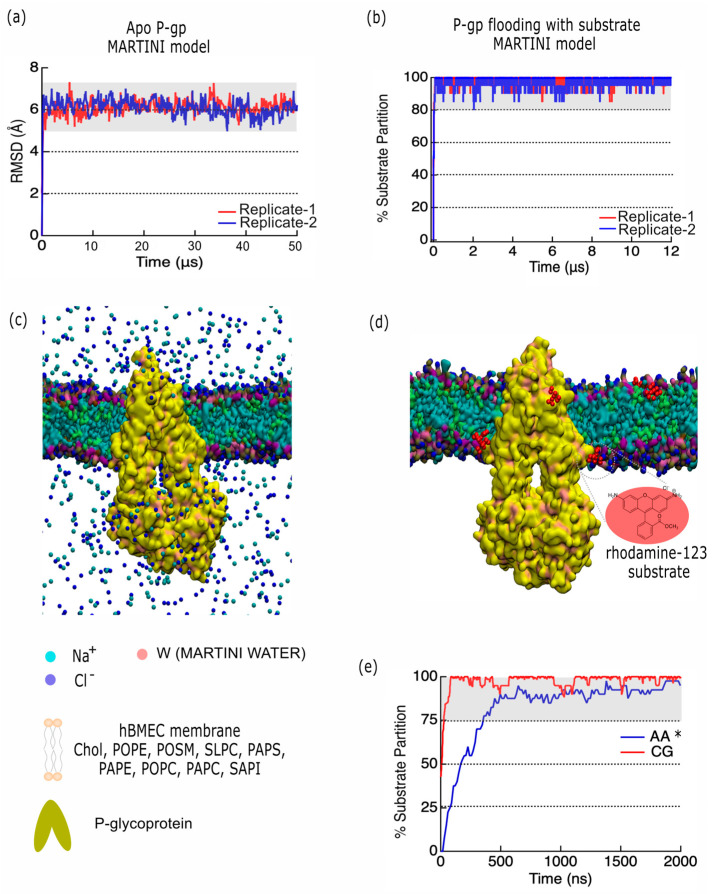
P-glycoprotein and P-gp-substrate coarse-grained models (*n* = 2 replicates). (**a**,**c**) Analysis of coarse-grained model of apo P-gp trajectory (50 μs) using MARTINI with an elastic network model [42]. The protein RMSD (μ ± σ) over 50 μs for replica 1 was found to be 6.1 ± 0.4 Å, and for replica 2 it was 6.1 ± 0.3 Å. (**b**,**d**) Partitioning of CG rhodamine 123 into the bilayer, defined as % of rhodamine 123 within 5 Å of the bilayer, using a MARTINI representation for replicate 1 (red) and 2 (blue). (**c**,**d**) System representations depicting P-gp (yellow) in a QuickSurf representation, with a BMEC membrane composed of 9 types of lipids. The ions are rendered in van der Waals shape. The W water bead is not rendered for visual clarity, but is given a legend representation as a red sphere. A legend of each component is given below, ion beads, water beads, membrane beads and the P-gp rendering. (**e**) Partitioning of CG rhodamine 123 into the bilayer, defined as % of rhodamine 123 within 5 Å of the bilayer, using a MARTINI representation (red) compared to the partitioning of atomistic rhodamine 123 from CHARMM General Force Field [44] (CGenFF) (blue) across 2 μs for purposes of equal comparison. Atomistic data, denoted by *, is sourced from our previous work [6]. © 2023 American Chemical Society. Figure panels were rendered with VMD 1.9.1 [34].

**Table 1 ijms-26-09050-t001:** Predicted IC_50_ and affinity probability obtained from Boltz-2 [37]. Summary of the confidence metrics for structural prediction. The predicted template modeling (pTM; 0.7–1: good; 0.5–0.7: average; 0–0.5: low) score considers the overall fold accuracy of a single protein chain [37]. The predicted inter-chain TM (ipTM; 0.8–1: good; 0.6–0.8: average; 0–0.6: low) score assesses the inter-chain interactions in multimeric models [37]. The confidence score (0.75–1: good; 0.5–0.75: average; 0–0.5 low) showcases the overall model quality. The average predicted local distance difference test (pLDDT; 0.7–1: good; 0.5–0.7: average; 0–0.5: low) is the confidence metric per residue [37] and reflects the local structural reliability. The affinity probability (0.75–1: good; 0.5–075 average; 0–0.5: low) estimates the likelihood that a predicted protein–ligand complex corresponds to a true, energetically favorable binding interaction [37]. The IC_50_ denotes the half-maximal inhibitory concentration, and is defined as the modulator concentration required to inhibit 50% of target activity, with a lower value denoting a higher modulator potency. The predicted negative log_10_ of IC_50_ (pIC_50_) correlates with the potency of inhibition, with a higher value predicting a greater inhibitory effect.

	Inhibitors			Substrates		
	QZ-Leu	Tariquidar	Elacridar	Taxol	BDE100	Ivacaftor
Predicted TM score (pTM)	0.763	0.772	0.781	0.780	0.792	0.774
Interface predicted TM score (ipTM)	0.890	0.911	0.933	0.892	0.779	0.915
Confidence score	0.743	0.785	0.784	0.755	0.781	0.777
Average predicted local distance difference test (pLDDT)	0.706	0.753	0.746	0.721	0.782	0.743
Affinity probability	0.402	0.667	0.672	0.545	0.503	0.663
Predicted pIC_50_	6.565	7.243	6.447	6.928	5.589	6.237
Predicted IC_50_ (nM)	272.3	57.1	357.3	118.0	2580	579.4

**Table 2 ijms-26-09050-t002:** Lipid distribution in BBB MARTINI 3.0.b.3.2 model. Values are provided in percentages and absolute lipid numbers. The total bilayer size is 322 lipids.

Lipid	Total Bilayer (%)
CHOL	30 (96)
POPE	6 (20)
POSM	19 (61)
SLPC	8 (27)
PAPS	8 (25)
PAPE	15 (47)
POPC	4 (13)
PAPC	8 (27)
SAPI	2 (6)

## Data Availability

The original data presented in the study are openly available for download on the GitHub webserver at https://github.com/chrisjorg/P-gp (accessed on 10 September 2025).

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
