# Peer review of "Simulation and Machine Learning Assessment of P-Glycoprotein Pharmacology in the Blood–Brain Barrier: Inhibition and Substrate Transport"

_ijms, 2025, doi:10.3390/ijms26189050_

Round 1
Reviewer 1 Report
Comments and Suggestions for Authors
This is a very interesting study on the role of the P-gp efflux pump in the development of multidrug resistance and the ability of P-gp substrates to permeate and transition across the blood–brain barrier. A significant background is presented to make clear the importance and need for conducting this study, modern methodology is used, and the results are very clearly presented and illustrated. I recommend publishing this article after minor corrections.
- In the section „Introduction“, it would be useful to add a brief comment on other AI studies of a P-gp efflux pump, if they are available, and what the advantage of this study would be compared to them.
- More keywords related to methodology should be added.
- Lines 40 and 44, the term (multidrug resistance) has been explained multiple times.
- The caption for Figures 2 - 5 should indicate which software was used for visualization. In addition, in subsection „4. Materials and Methods“ the version of the software listed should be indicated.
- In model quality evaluation, since the SWISS-MODEL workflow did not retain the original ligand-binding sites and the ligands were manually reinserted, what confirms the validity of the protocol?
- It would be useful to highlight possibilities and future perspectives of application of the results obtained in this study in the section „Conclusion“.
Author Response
Comment 1: In the section „Introduction“, it would be useful to add a brief comment on other AI studies of a P-gp efflux pump, if they are available, and what the advantage of this study would be compared to them.
Response 1: We thank the reviewer for this insightful comment. We have added an additional paragraph describing other AI studies of P-gp efflux (page 3, line 89):
“In addition to atomistic simulations, the in silico repertoire includes a rising number of AI and machine learning (ML) tools applied to predict novel P-gp substrates and inhibitors. Such studies have used a broad range of techniques, and we highlight just a few. Supervised ML models such as Support Vector Machines (SVM) [27–29] were used since the early 2000s to predict novel P-gp substrates, and with great success. As a matter of fact, the often quoted number of 300 P-gp substrates, originates from a paper that utilised SVMs [29]. Other papers have used convolutional neural networks [30], such as the Novel Ligand-based Convolutional Neural Network (NLCNN) model that classfies P-gp substrates to a prediction accuracy of 80%, based on a curated dataset of 197 P-gp substrates, by integrating molecular docking and ligand-based deep learning methods for further predictive improvement [30]. Similarly, graph neural networks (GNNs), such as the AttentiveFP model [31] that is trained on a dataset of 1995 drug molecules (1202 substrates, 793 nonsubstrates) and has a Receiver Operating Characteristic Area Under the Curve (ROC-AUC) of 0.848 and an accuracy of 0.815.”
- Rehman, M.U.; to Chong, K.; Tayara, H. Drugs Inhibition Prediction in P-Gp Enzyme: A Comparative Study of Machine Learning and Graph Neural Network. Computational Toxicology 2025, 100344.
- Doniger, S.; Hofmann, T.; Yeh, J. Predicting CNS Permeability of Drug Molecules: Comparison of Neural Network and Support Vector Machine Algorithms. Journal of computational biology 2002, 9, 849–864.
- Xue, Y.; Yap, C.W.; Sun, L.Z.; Cao, Z.W.; Wang, J.F.; Chen, Y.Z. Prediction of P-Glycoprotein Substrates by a Support Vector Machine Approach. J Chem Inf Comput Sci 2004, 44, 1497–1505.
- Neela, M.M.V.; Peramss, S.R. A Novel Ligand-Based Convolutional Neural Network for Identification of P-Glycoprotein Ligands in Drug Discovery. Mol Divers 2025, 1–18.
- Hsu, K.-C.; Wang, P.-H.; Su, B.-H.; Tseng, Y.J. A Robust and Interpretable Graph Neural Network-Based Protocol for Predicting p-Glycoprotein Substrates. Brief Bioinform 2025, 26, bbaf392.
Comment 2: More keywords related to methodology should be added.
Response 2: We thank the reviewer for this insightful comment. We have updated the keywords section (page 1, line 29):
Keywords: blood-brain barrier, efflux pumps, P-glycoprotein, molecular dynamics, cryo-EM, substrates, inhibitors, PDB database mining, coarse-grain modelling, Boltz-2.
Comment 3: Lines 40 and 44, the term (multidrug resistance) has been explained multiple times.
Response 3: We thank the reviewer for this comment. This has been corrected to only be stated once.
Comment 4: The caption for Figures 2 - 5 should indicate which software was used for visualization. In addition, in subsection „4. Materials and Methods“ the version of the software listed should be indicated.
Response 4: We thank the reviewer for this comment. We have added a clarification to figures 2 to 5:
Figure 2: “Figure panels were rendered with VMD 1.9.1 [34]”
Figure 3: “Figure panels were rendered with VMD 1.9.1 [34]”
Figure 4: “Figure panels were rendered with VMD 1.9.1 [34]”
Figure 5: “Figure panels were rendered with VMD 1.9.1 [34]”
We have added a paragraph to the Materials and Methods detailing the software used (page 17, line 491):
“4.5 Software
All chemical renderings were prepared with VMD 1.9.1 [34]. Simulations were run with the GROMACS 2021 simulation package [55]. Final figures were prepared with Inkscape.”
Comment 5: In model quality evaluation, since the SWISS-MODEL workflow did not retain the original ligand-binding sites and the ligands were manually reinserted, what confirms the validity of the protocol?
Response 5: We thank the reviewer for this important question. We tested both inputting the ligand into the homology modelling (HM) procedure, as well as re-adding after using a strict STAMP structural alignment procedure between template and the resulting HM (see Figure 1 below). This alignment showed that the ligand position was not affected. We are confident in this approach.
Figure 1. STAMP (Structural Alignment of Multiple Proteins) structural alignment of the original mouse template to the resulting human homology, using MultiSeq [54] in VMD [34]. The substrate from the original template is depicted together with the reinserted pose. The respective templates are (A) PDB id 6UJN; X-ray, 3.98 Å) [35], and (B) PDB id 7OTG; cryo-EM, 5.40 Å) [36].
We have added the following clarification to the main text (page 16, line 452):
“STAMP (Structural Alignment of Multiple Proteins) structural alignment of template and resulting homology model was done with MultiSeq [54] in VMD [34] prior to reinsertion.”
Comment 6: It would be useful to highlight possibilities and future perspectives of application of the results obtained in this study in the section „Conclusion“.
Response 6: We thank the reviewer for this comment. We have addressed this in the Conclusions section (page 15, line 423):
“The P-gp multimodal model holds promise to help advance our clinical knowledge of P-gp inhibition. When considering that all prior clinical trials P-gp inhibition have been classed as unsuccessful at the late stage [50], this challenge thus becomes more urgent. Our P-gp coarse-grain model could be used for future studies to better understand how P-gp substrates interact with the protein at long timescales, although the main bottleneck is the non-trivial nature of generating small-molecule simulation parameters.”

Reviewer 2 Report
Comments and Suggestions for Authors
This work addresses a significant challenge in CNS disease treatment, that is, the presence of efflux transporters like P-gp largely limits the CNS delivery of drugs. The results provide insights into the pharmacology of P-gp and its binding to inhibitors and substrates. Here are some suggestions.
Abstract line 25: The expression "substrate permeation into P-gp" may be unclear. Do the authors mean the permeation of the substrate across the cell membrane that expresses P-gp?
Figure 6: Please clarify the difference between panel c and panel d, and the distinct information each panel is intended to convey in the figure legend.
Line 357: The Boltz-2-predicted IC50 for QZ-Leu and QZ-Phe were quite different from reported estimations. Could you comment on/discuss about this?
Line 361: The expression of "sub-nanomolar efficacy (Kd = 5.1 nM)" is not accurate. Kd is a measurement of binding affinity, not efficacy.
Table 2: What is the unit for the absolute values?
Author Response
Comment 1: Abstract line 25: The expression "substrate permeation into P-gp" may be unclear. Do the authors mean the permeation of the substrate across the cell membrane that expresses P-gp?
Response 1: We thank the reviewer for this comment. We have clarified the statement as (page 1, line 26):
“Our coarse-grain model of substrate permeation into membranes expressing P-gp shows benchmarking similarities to prior atomistic models and provide new insights at far longer timescales.“
Comment 2: Figure 6: Please clarify the difference between panel c and panel d, and the distinct information each panel is intended to convey in the figure legend.
Response 2: We thank the reviewer for this comment. Panel (c) denotes the CG model of apo P-gp, in which no substrate is populated in the box. Panel (d) denotes the CG model of holo P-gp, in which n substrate molecules are populated in the box.
We wish to clarify, all left panels (a, c) correspond to a simulation of apo P-gp, in which no substrate is populated in the box. All right panels (b, d, e) correspond to holo P-gp, in which n substrate molecules are populated in the box.
Panels (a): The protein root mean-square deviation (RMSD) provides the structural variation from the initial conformation throughout the simulation. This is a measure of structural stability. The structure deviated 6.1 ± 0.4 Å from the initial conformation in simulation replicate-1, and similarly, 6.1 ± 0.3 Å in replicate-2.
Panel (b): The % of substrate molecules that have partitioned into the membrane (within 5 Å of the bilayer).
Panels (c-d): Depicts a snapshot of the P-gp simulation box either in the absence (c) or presence (d) of substrate molecule in the simulation box. Panel (c) depicts the setup of the coarse-grain model. Panel (d) in addition shows the partitioning of the substrate molecule into the membrane.
Panels (e): Shows the comparison of the % partitioning for the same substrate (named rhodamine-123) between a coarse-grain simulation replicate and an all-atom MD simulation from [6].
Comment 3: Line 357: The Boltz-2-predicted IC50 for QZ-Leu and QZ-Phe were quite different from reported estimations. Could you comment on/discuss about this?
Response 3: We thank the reviewer for raising this interesting point.
The original text read: “A trend emerges whereby the earlier first-generation inhibitors such as QZ-Leu [7] are associated with less potent P-gp inhibition, and larger IC50 values. For QZ-Leu, and the associated QZ family of inhibitors, Szewczyk et al. estimate IC50 values in vitro to increase with the size of the side-chain, namely QZ-Val (IC50 = 1.7 µM), QZ-Leu (IC50 = 5.4 µM) and QZ-Phe (IC50 = 24 µM) [7]. We proceed to compare these experimental values to those estimated by Boltz-2, namely QZ-Leu (IC50 = 0.27 µM) and QZ-Phe (IC50 = 0.26 µM) (Table S5).”
We proceeded to use Boltz-2 to calculate the IC50 for QZ-Val, which was missing, and have revised the text (page 13, line 391):
“Szewczyk et al. estimate IC50 values in vitro to increase with the size of the side-chain, namely QZ-Val (IC50 = 1.7 µM), QZ-Leu (IC50 = 5.4 µM) and QZ-Phe (IC50 = 24 µM) [7]. We proceed to compare these experimental values to those estimated by Boltz-2, namely QZ-Val (IC50 = 1.4 µM), QZ-Leu (IC50 = 0.27 µM) and QZ-Phe (IC50 = 0.26 µM) (Table S5). The ratio between Boltz-2 IC50 values to the experimental IC50 (IC50,Boltz/IC50,invitro) are as follows: QZ-Val: ~1 QZ-Leu ~20 QZ-Phe ~92.”
We clarify, the ratio between Boltz-2 IC50 values to the experimental IC50 (IC50,Boltz/IC50,invitro) are:
QZ-Val: ~1
QZ-Leu ~20
QZ-Phe ~92
We have added these ratios to the manuscript.
We agree that compared to the strong agreement for tariquidar (experimental IC50 of 43 nM compared to an IC50 of 57 nM from Boltz-2), the deviations are much larger for the QZ variants, especially with the bulkier side chain Phe. This could arise from the absence of QZ-X (X=Phe) variants in the training sets used by Boltz-2, or could be due to inherent methodological differences in the experimental estimation.
Comment 4: Line 361: The expression of "sub-nanomolar efficacy (Kd = 5.1 nM)" is not accurate. Kd is a measurement of binding affinity, not efficacy.
Response 4: We thank the reviewer for requiring clarification. We agree the language was not clear. We have restated this and added an experimental IC50 value of tariquidar to P-gp (page 13, line 395):
“For the tariquidar dimer (Figure 2.b), Boltz-2 predicts the IC50 of tariquidar in the nanomolar regime (IC50 = 57.1 nM). This is in agreement with experimental characterization of tariquidar as a potent third-generation P-gp inhibitor (IC50 = 43 nM [49]) that binds strongly to P-gp (Kd = 5.1 nM) [50]. “
Comment 5: Table 2: What is the unit for the absolute values?
Response 5: We thank the reviewer for requesting this clarification. We have added the following text to the manuscript (page 14, line 476):
“Values are provided in percentage and (absolute lipid number) values. The total bilayer size is 322 lipids.”
